# *Monascus* Red Pigment Liposomes: Microstructural Characteristics, Stability, and Anticancer Activity

**DOI:** 10.3390/foods12030447

**Published:** 2023-01-18

**Authors:** Pengcheng Long, Lisha Zhu, Huafa Lai, Suyin Xu, Xingxing Dong, Yanchun Shao, Liling Wang, Shuiyuan Cheng, Gang Liu, Jingren He, Yi He

**Affiliations:** 1National R&D Center for Se-Rich Agricultural Products Processing, Hubei Engineering Research Center for Deep Processing of Green Se-Rich Agricultural Products, School of Modern Industry for Selenium Science and Engineering, Wuhan Polytechnic University, Wuhan 430023, China; 2Key Laboratory for Deep Processing of Major Grain and Oil, Ministry of Education, Hubei Key Laboratory for Processing and Transformation of Agricultural Products, School of Food Science and Engineering, Wuhan Polytechnic University, Wuhan 430023, China; 3College of Food Science and Technology, Huazhong Agricultural University, Wuhan 430070, China; 4College of Life Science, Tarim University, Alar 843300, China

**Keywords:** *Monascus* red pigments, liposomes, morphological characteristics, stability, anticancer activity

## Abstract

*Monascus* red pigments (MRPs), which are a kind of natural colorant produced by *Monascus* spp., are widely used in the food and health supplements industry but are not very stable during processing and storage. Thus, MRPs were embedded into liposome membranes using a thin-film ultrasonic method to improve stability in this study. *Monascus* red pigments liposomes (MRPL) exhibited spherical unilamellar vesicles (UV) with particle size, polydispersity indexes (PDI), and zeta potential of 20–200 nm, 0.362 ± 0.023, and −42.37 ± 0.21 mV, respectively. pH, thermal, light, metal ion, storage, and in vitro simulated gastrointestinal digestion stability revealed that, compared with free MRPs, liposomes embedding significantly enhanced the stability of MRPs when exposed to adverse environmental conditions. Furthermore, anticancer assay suggested that MRPL exhibited a stronger inhibitory effect on MKN-28 cells by damaging the integrity of cells, with the IC_50_ value at 0.57 mg/mL. Overall, MRPLs possess stronger stability in external environment and in vitro simulated digestion with greater anticancer activity, indicating that MRPLs have the potential for promising application in the functional foods and pharmaceutical industries.

## 1. Introduction

*Monascus* pigments (MPs), a kind of natural edible pigment, are the mixture of azaphilones consisting of yellow, orange, and red pigments produced by *Monascus* strains [1]. MPs have gained wide acceptance in Southeast Asian areas, typically in China, Japan and Indonesia, for their bright colors, good solubility, and highly biosafety [2]. Traditionally, as safety colorants, MPs are not only added to various meat products, baked foods, and condiments, but also serve as food preservatives to prolong shelf life. Lots of reports emphasized that MPs have been deemed as a substitute for nitrate and nitrite salts that are harmful to human beings [1,3]. Among MPs, MRPs have a promisingly application because red can be added into various foods, and natural red pigments that are safe and suitable for food are difficult to access [4]. MRPs have the characteristic of good solubility in water and alcohol. Moreover, MRPs possess lots of beneficial biological and therapeutic activities, such as anticancer [5], antioxidant [3], and anti-obesity [6] activities. However, MRPs are sensitivity to some environmental factors, such as pH, high temperature, oxidation, enzymes, metal ions, and light [7,8]. Therefore, MRPs exhibit instability, degradation, and discoloration during storage and processing, which largely increases the limitation in practical application.

So far, there are so many ways to improve the stability of natural pigments. Zhang et al. [9] reported that microencapsulated anthocyanins prepared by emulsification/internal gelation have higher stability, with the lowest degradation constant and longest half-time compared with free anthocyanins. Shaddel et al. [10] confirmed that the stability of black raspberry anthocyanins encapsulated using gelatin and gum Arabic increased up to 23.66% after 60 days of storage. Moreover, MPs’ double emulsions exhibited good stability against heating treatment (90 °C, 30 min) due to the increased interaction between the droplets [8]. Liposomes, a promising carrier of hydrophobic and hydrophilic compounds, are spherical vesicles that consisted of phospholipids and cholesterol [11]. Based on the diameter size and structure, liposomes are dived into three major groups: small unilamellar vesicle (SUV) (diameter of 25–100 nm), large unilamellar vesicle (LUV) (diameter exceed 100 nm), and a multilamellar vesicle (MLV) (diameter of 500–5000 nm) [12]. More importantly, liposomes possess many beneficial properties, such as low toxicity, good biocompatibility, and target specificity [13]. Therefore, liposomes embedding has been widely used, not only in drug delivery, but also in food, cosmetics, and agriculture areas, to protect target products from being degraded [14]. Caddeo et al. [15] reported that PEG-modified liposomes efficiently preserved the bioactivity of resveratrol when delivered in the biological fluids.

Compared with other encapsulation technologies, such as microcapsules, coacervation and emulsion, one of the main advantages of liposomes is that they can typically provide stability in high-water containing foods. Furthermore, the limitation of applications in food systems may be reduced because of the natural encapsulation materials (phospholipids and cholesterol) [16]. The liposome system has been widely employed to slow the release of core materials and improve bioavailability in vivo [12,17]. Thus, the aim of this study is to prepare MRPs liposomes by a thin-film ultrasonic method to improve their stability. The stability of MRPL was evaluated in different conditions of pH, thermal, light, metal ion, storage, and in vitro simulated gastrointestinal digestion. Moreover, the cytotoxicity of MRPL to MKN-28 and HepG-2 cells were also studied. This study has made a systematical assessment of MRPL on microstructure characteristics, stability, in vitro release and cell cytotoxicity. 

## 2. Materials and Methods

### 2.1. Materials

MRPs were purchased from Yunnan Tonghai Yang Natural Products Co., Ltd. (Yunnan, China). Soybean lecithin and cholesterol were obtained from Henan Huayue Chemical Products Co., Ltd. (Henan, China) and Chengdu Kelong Chemical Reagent Co., Ltd. (Chengdu, China), respectively. Pepsin (1:10,000) and trypsin (1: 250) were purchased from Shanghai Solarbio Biotechnologr Co., Ltd. (Shanghai, China). Roswell Park Memorial Institute (RPMI) 1640 medium, Dulbecco’s modified eagle medium (DMEM) and fetal bovine serum were purchased from Thermo Fisher Scientific Inc. (Waltham, MA, USA). Penicillin, streptomycin and trichloroacetic acid solution were provided by Sigma-Aldrich Trading Co., Ltd. (Shanghai, China). Cell Counting Kit-8 (CCK8) was purchased from Beyotime Biotechnology Inc. (Shanghai, China). MKN-28 and HepG-2 cells, derived from human colon adenocarcinoma and hepatoma, respectively, were bought from BioWit Technologies Co., Ltd. (Shenzhen, China). All chemicals used in this study were of analytical grade.

### 2.2. Preparation of MRPL

MRPL was prepared by the thin-film ultrasonic method described by Isailović et al. [16] with moderate modification. In brief, 1 g of soybean lecithin and 0.2 g of ultra-pure cholesterol were dissolved in 100 mL of chloroform, and 4 mg of MRPs was mixed subsequently. The mixture was transferred into a 250 mL round flask to completely remove the chloroform, using a R-3 rotary vacuum evaporator (rotate at 100 rpm) (Buchi Laboratory Equipment Trading Co., Ltd., Shanghai, China) at 37 °C. After the thin lipid film was formed, 100 mL of distilled water was added to hydrate with the lipid film by rotating for 30 min at 200 rpm in a 37 °C water-bath. After that, the solution was treated with ultrasound (JY92-IIN, biotechnology Co., Ltd., Ningbo, China) at 400 W for 4 min with 3 s pulse-on and 3 s pulse-off in an ice-bath to transform liposomes from MLV to UV, and then successively filtered through 0.45 μm and 0.22 μm water microfiltration membranes to obtain MRPL suspension [12,18]. The suspension was centrifugated at 45,000 rpm for 30 min at 4 °C using a Sorvall MTX 150 ultracentrifuge (Thermo Fisher Scientific Inc., Waltham, MA, USA) to collected MRPL precipitate and to remove the unencapsulated MRPs; the encapsulation rate of MRPs was 49% ± 2.6%.

### 2.3. Characterization of MRPL

#### 2.3.1. Microstructural Characteristic

The morphology of MRPL was observed using a JEM-1000 transmission electron microscope (TEM) (HITACHI Co., Ltd., Tokyo, Japan) at a voltage of 200 kV. Briefly, 1% (*w*/*v*) phosphotungstic acid (PTA) was used to negative-stain MRPL in a pH range of 6.5 to 7.0 for 30 s, then a drop of MRPL was placed on the copper grid and dried naturally at room temperature. After that, the microstructure of MRPL was photographed with 100,000 times magnification.

#### 2.3.2. PDI and Zeta Potential

The prepared liposomes suspension was diluted 100 times with distilled water, then PDI and zeta potential were measured at 25 °C using a Nano ZS90 Particle Sizer (Malvern Instruments Ltd., Worcestershire, UK).

### 2.4. Stability of MRPL

#### 2.4.1. Degradation Rate

The standard curve of the MRPs was obtained by measuring the maximum absorbance (500 nm) of the MRPs water solution at different concentration gradients. The regression equation between the MRPs solution concentration and the absorbance was described as following, with correlation coefficient (R^2^) as 0.9999:Y = 8.615X − 0.0059(1)

Based on the standard MRPs solution curve, the initial concentration of MRPs in the MRPL was calculated after adding 10% (*v*/*v*) Triton X-100 to break the membrane [19]. After different treatments, the solutions were centrifugated at 45,000 rpm for 30 min at 4 °C using a Sorvall MTX 150 ultracentrifuge (Thermo Fisher Scientific Inc., Waltham, MA, USA) to collected MRPL precipitate [16]. The precipitate was re-dissolved in distilled water which contained 10% (*v*/*v*) Triton X-100 and mixed using magnetic stirring (Gongyi Instrument Co., Ltd., Gongyi, China), and the wavelength was measured at 500 nm using an ultraviolet-visible spectrophotometer (Wuhan Jiejing Scientific Instrument Co., Ltd., Wuhan, China). The degradation rate (DR) of MRPL after various treatments was calculated using Equation (2) [9]. As control, a free MRPs solution of the same concentration was treated under the same conditions; the calculation of DR of free MRPs was the same as MRPL.
(2)$$\mathrm{DR}(\%)=\frac{C_{0}-C_{t}}{C_{0}}\times 100$$
where C_t_ is the content of MRPs during different treatments, and C_0_ is the initial content of MRPs.

#### 2.4.2. pH Stability

To compare the pH stability, the pH of MRPL and MRPs solution with the same concentration was adjusted to 2, 3, 4, 5, 6, 7, 8, 9, and 10 using 0.1 M NaOH or HCl solution and stored in the dark for 5 h at room temperature. The DR of free MRPs and MRPL were calculated using Equation (2), respectively. 

#### 2.4.3. Temperature Stability

The solutions of free MRPs and MRPL were heated in a water bath at 60 °C, 70 °C, 80 °C, and 90 °C for 5 h along with being kept in the dark, and the DR of free MRPs and MRPL were calculated using Equation (2), respectively.

#### 2.4.4. Light Stability

The solutions of free MRPs and MRPL were exposed to illumination (500 lx) for 0 to 4 h at room temperature with the other conditions unchanged, and the DR of free MRPs and MRPL at every 30 min interval was calculated using Equation (2), respectively.

#### 2.4.5. Metal Ions Stability

The solutions of free MRPs and MRPL were mixed with the same volume of 0.1 mol/L NaCl, KCl, CaCl_2_, ZnSO_4_, FeSO_4_, and CuSO_4_ solutions and stored in the dark at room temperature for 24 h, and the DR of free MRPs and MRPL was calculated using Equation (2), respectively.

#### 2.4.6. Storage Stability

The solutions of free MRPs and MRPL were stored in refrigeration at 4 °C for 0, 2, 4, 6, 8, 10, 15, 20, 25, or 30 days in the dark, and the DR of free MRPs and MRPL were calculated using Equation (2), respectively.

### 2.5. Stimulated Gastrointestinal Digestion In Vitro

For gastric condition, 100 mL of simulated gastric fluid (SGF) was obtained by adding 1 g pepsin enzyme and 0.2 g NaCl with the pH value adjusted to 1.2 using 0.1 M HCl. The suspensions of free MRPs and MRPL of the same concentration were added to the simulated gastric fluid, respectively, and placed in a shaking bed at a temperature of 37 °C and a rotation speed of 100 rpm for 5 h. 

For intestinal condition, 100 mL of simulated intestinal fluid (SIF) was obtained by adding 1 g of trypsin and 0.68 g of KH_2_PO_4_, and the pH values were adjusted to 6.8 using 0.1 M NaOH solution. The suspensions of free MRPs and MRPL of the same concentration were added to the simulated intestinal fluid, respectively, and incubated for 5 h at the same condition as gastric digestion. The DR of free MRPs and MRPL was calculated using Equation (2), respectively.

The release behaviors of MRPs and MRPL in a simulated gastrointestinal environment were analyzed by four drug-release kinetic models [20,21]:

Zero order model: (3)$$\frac{C_{0}-C_{t}}{C_{0}}=\mathrm{Kt}$$

First order model:(4)$$\mathrm{In}\frac{C_{t}}{C_{0}}=-\mathrm{Kt}$$

Higuchi model: (5)$$\frac{C_{0}-C_{t}}{C_{0}}={\mathrm{Kt}}^{0.5}$$

Korsmeyer–Peppas model:(6)$$\frac{C_{0}-C_{t}}{C_{0}}={\mathrm{Kt}}^{n}$$
where K is the kinetic constant and n describes the kinetic index.

### 2.6. Anticancer Activity to MKN-28 and HepG-2 Cells

#### 2.6.1. Cells Culture

HepG-2 and MKN-28 cells were cultured in high-glucose RPMI1640 and DMEM medium, respectively, and supplemented with 10 mL/100 mL inactivated fetal bovine serum and 1 mL/100 mL double-antibiotics (penicillin and streptomycin). The cells were subcultured in a CO_2_ incubator maintained at 5% CO_2_ at 37 °C for 2–3 generations.

#### 2.6.2. Cells Cytotoxicity

The cells’ viability was measured by the method described by Ren et al. [22] with slight modification. Five different concentrations of MRPs and MRPL suspension (0.02, 0.04, 0.08, 0.16, and 0.32 mg/mL) were filtered by 0.22 μm filter to remove bacteria. When the cells hit their logarithmic stage, 100 μL of cells suspension (1.5 × 10^5^ cells/mL) were seeded in 96-well plates and cultured in an incubator for 12 h. Thereafter, the same volume of MRPs and MRPL solution were added into HepG-2 and MKN-28 cells suspension after replacing the medium, and cultured for 2 h in a CO_2_ incubator at 37 °C. The groups without any MRPS or MRPL treatment were control groups. Subsequently, 10 μL of CCK-8 solution was added for incubating for 4 h, and then the absorbance value at 450 nm was measured with an auto ELISA detector (iMark, Bio-Rad, Hercules, CA, USA). The inhibition rates (IR) of MRPs and MRPL to HepG-2 and MKN-28 cells were calculated using Equation (7):
(7)$$\mathrm{IR}(\%)=[1-\frac{{\mathrm{OD}}_{1}-{\mathrm{OD}}_{2}}{{\mathrm{OD}}_{3}-{\mathrm{OD}}_{4}}]\times 100$$
where OD_1_ is the absorbance value of the sample group, OD_2_ is the absorbance value of the medium which contained CCK-8 solution, and OD_3_ is the absorbance value of the control group.

The values of IC_50_ were calculated based on IR and logarithmic concentrations of MRPs and MRPL [23].

#### 2.6.3. Cells Morphology

The cells were washed three times with 2 mL phosphate-buffered solution (PBS) buffer for 30 s to remove external medium and 1 mL Hank’s balance salt solution (HBSS) was added after treatment with or without 0.32 mg/mL MRPs or MRPL, as above; subsequently, the cells’ morphology was observed using a SDPTOP CX40M microscope (Sunny Optical Technology (Group) Co., Ltd., Zhejiang, China) at 200 times magnification.

### 2.7. Statistical Analysis

All experiments were performed in triplicate and data were reported as the mean values ± standard deviations (SD). Statistical analysis was performed using the software Statistical 8.0 (StatSoft Inc., Tulsa, OK, USA), and the independent samples t-test and one-way analysis of variance (ANOVA) with Tukey’s post-test at 95% probability were used to identify differences among different groups. *p* < 0.05 was considered statistically significant and marked by *; *p* < 0.01 was considered extremely significant and marked by **.

## 3. Results and Discussion

### 3.1. Particle Size and Morphology

The TEM image exhibited the morphology, distribution and particle size of MRPL, as shown in Figure 1. MRPL appeared to be a predominantly spherical UV with even distribution, and the particles’ size varied from 20 to 200 nm with small vesicles occupying the majority, which is consistent with previous studies that reported that nanoliposomes possess a morphology of spherical vesicles along with a particle size varying from tens to hundreds [11,24,25]. Some reports emphasized the bigger particle size of liposomes having a positive connection with the increasing cholesterol content [16,26]. The particle size is also influenced by prepared methods, such as extrusion and sonication technique [16]. Song et al. [17] confirmed that the size and distribution were influenced by saturations of phospholipids; liposomes made of soybean phosphatidylcholine exhibited a minimum average size compared with those made of egg yolk phosphatidylcholine and hydrogenated soy phosphatidylcholine. The microspheres smaller than 100 μm not only exhibit great stability and solubility [27], but also contribute to delivering bioactive compounds to various sites in vivo [28]. Nanoliposomes that have particle sizes smaller than 200 nm have high efficacy and utilization in vivo, such as improving vascular permeability and decreasing the chance of uptake by the reticuloendothelial system [29], which indicates that MRPL has the potential of delivering in vivo because of its small particle size.

### 3.2. PDI and Zeta Potential

The PDI and zeta potential of MRPL were 0.362 ± 0.023 and −42.37 ± 0.21 mV, respectively. In general, the value of PDI changing from 0 to 1.0 represents the distribution of particle size transforming from homogeneous to heterogeneous [30]. Isailović et al. [16] reported that the value of PDI was largely influenced by the processing methods, and the sonicated resveratrol liposomes (PDI from 0.15 to 0.22) had larger PDI value than extruded samples (PDI from 0.23 to 0.47). Moreover, zeta potential is the electrostatic charge of the liposome and that of MRPL was greater than 30 with a negative charge, manifesting that the repulsive force can efficiently protect MRPL against aggregation to enhance the stability of the MRPL system [11,26]. In the previous study, anthocyanin liposomes that were produced by supercritical carbon dioxide exhibited a particle size of 159.3 ± 0.16 nm, PDI of 0.244 ± 0.02 and zeta potential of −40.2 mV [31], which was largely consistent with the results of MRPL. Lu et al. [24] reported that the encapsulation of allicin into liposomes significantly improved its storage stability, and the mean size as well as zeta potential of allicin nanoliposomes was 145.27 ± 15.19 nm and −40.10 ± 0.96 mV, respectively. 

### 3.3. Stability of MRPL

#### 3.3.1. pH Stability

As shown in Figure 2a, both MRPs and MRPL were unstable in a heavily acidic or alkaline environment such as pH ≤ 4 and pH ≥ 8, with the DR of MRPs and MRPL increased to approximately 40–50%. The hydrolysis constant of lecithin increased when the pH was greater than or less than its isoelectric point (pI) (6.5), which means that strong acid or alkali can destroy the structure of liposomes, resulting in holes and polymerization of the liposome membrane, and reducing liposome stability [32,33]. However, the DR of MRPL was significantly lower than that of MRPs, indicating that wrapping by lipids could successfully enhance the stability of MRPs in adverse conditions. Moreover, both MRPs and MRPL exhibited better stability in slightly acidic or near neutral environments. For example, the DR of MRPL at pH 5 was 20.6%, which was 9.6% lower than that of MRPs; the DR of MRPL at pH 6 was 16.3%, which was 3.45% lower than that of MRPs. Chi et al. [14] reported that anthocyanins that were stable under acidic conditions had a retention rate of 40.8%, while that of anthocyanins nanoliposomes (AN) was 60.6% when exposed to pH 7, indicating that double-layer phospholipid membranes of liposomes can usefully prevent core materials from degradation.

#### 3.3.2. Temperature Stability

As shown in Figure 2b, the DR of MRPs and MRPL continued to increase with the increasing of the temperature, indicating that the stability of MRPs and MRPL is susceptible to high temperature [34]. The DR of MRPL was about 30% after being heated at 60 °C for 5 h, which was 77% of that of MRPs. When the temperature increased from 60 °C to 90 °C, the DR of MRPL went up to 78.2%, which was nearly 2.6-fold higher than that at 60 °C, indicating that the structure of MRPL was significantly impaired by high temperature. However, there was a significant difference (*p* < 0.05) between MRPL and MRPs on DR at 90 °C: the DR of MRPL was 12.2% lower than that of MRPs. Jung et al. [35] reported that about 29% of MRPs remained after storage for 30 days at 30 °C, while there were no pigments remaining in the same time span at 80 °C, and the half-time of MRPs at 80 °C was 3.5 days. Overall, compared with free MRPs, although liposome embedding helped to prevent MRPs from being degraded at temperatures ranging from 60 °C to 90 °C, high temperature accelerated oxidation and breakage of the lipid bilayer of MRPL [36]. Therefore, neither MRPs nor MRPL are suitable for processing at high temperature.

#### 3.3.3. Light Stability

As shown in Figure 2c, the DR of MRPs rapidly went up from 1.9% to 12.9%, while that of MRPL increased from 1.5% to 8.6%, when the time exposed to illumination (500 lx) increased from 0.5 to 2 h. With the extension of exposure time, the degradation had a tendency of slowly increasing. The DR of MRPs and MRPL rose to 18.4% and 13.2%, respectively, during 4 h illumination. Jung et al. [35] claimed that the color of MRPs exposed to sunlight (intensity of 2.33 ± 0.26 MJ/m^2^) for 18 h changed to brown with a weaker red, which was mainly caused by the decomposition or combinations of new compounds in MRPs. In general, statistical analyses revealed that there was a significant difference (*p* < 0.05) in the degradation of MRPs and MRPL after exposure to 4 h illumination, indicating that liposome embedding efficiently protected MRPs from being degraded under light of 500 lx. Chi et al. [14] reported that AN has lower degradation than free ones when treated with white light, which is outlined with the results of this study. 

#### 3.3.4. Metal Ions Stability

As shown in Figure 2d, MRPs and MRPL exhibited the least stability in FeSO_4_ solution, with the DR of MRPs and MRPL at 21.3% and 14.2%, respectively. The instability is partly because Fe^2+^ was at a state of readily being oxidized or reduced, and it was easy to have complexation with MRPs, which leads to a color change of the solution, suggesting that there is little benefit to preserving MRPs or MRPL in an iron container. Moreover, the DR of MRPL in a Ca^2+^, Cu^2+^ or Zn^2+^ environment was 11.9%, 14.2%, and 9.8%, which were 4.3%, 7.1%, and 5% lower than those of MRPs, respectively. Moreover, both MRPs and MRPL displayed good stability in NaCl and KCl solutions, typically in the KCl solution. Anthocyanins, a kind of natural edible pigment, have the characteristic of easily forming metal-anthocyanin complexes because of the existence of o-di-hydroxyl groups in the B ring [37]. Thus, the groups in the skeleton of polyketide of MRPs may contribute to the complexation with some metal ions. Statistical analysis revealed that there were extremely significant differences (*p* < 0.01) in DR between MRPs and MRPL in six metal ions solutions, indicating that MRPLs possess a protective effect on MRPs when exposed to metal ions solutions.

#### 3.3.5. Storage Stability

Figure 2e shows the DR of MRPs and MRPL at 4 °C for 30 days’ storage in the dark. Although the DR increased as the storage duration increased, the degradation kept at a small rate with lower than 10%. Dark treatment at 4 °C for 30 days resulted in 9.9% and 7.1% degradation for free MRPs and MRPL, respectively. Statistical analyses revealed that there was a significant difference (*p* < 0.05) between the DR of MRPs and MRPL after long time storage, which was consistent with the result of Tai et al. [38] that all chitosan-coated, curcumin-loaded liposomes displayed enhanced stability resistant to long-term storage at 4 °C. After long-term storage, the poor stability of the liposomes was attributed to aggregation, fusion, oxidation, and the hydrolysis of phospholipids, which led to the destruction of their integrity and leakage of the encapsulated material [36].

#### 3.3.6. In Vitro Release

The DR of MRPs and MRPL in simulated gastrointestinal environment is shown in Figure 3. As can be seen in Figure 3a, the DR of MRPs and MRPL in SGF dramatically went up as the time increased. After 2 h treatment at pH 1.2, the DR of MRPs and MRPL was approximately 28.6% and 22.1%; then, the rate increased to 65.6% and 55.4%, respectively, when treated for 5 h. The trend of degradation was lined with the results of pH stability that both MRPs and MRPL were more likely to degrade in strong acid environment (pH 1.2). Owing to the existence of pepsin, the DR of MRPs and MRPL in SGF was greater than that in mere strong acid environments. Moreover, the degradation of MRPs and MRPL was faster in intestinal environments than in gastric environment, as shown in Figure 3b, which is outlined by the report of Chi et al. [14]. The DR of MRPs in SIF was about 50.3% after 2 h treatment at pH 6.8, which was one-fold higher than that of MRPL. After 5 h treatment, the DR of MRPs and MRPL was approximately 80.2% and 60.5%, respectively, indicating that the slower release rate of MRPL benefited from the protection of liposome embedding [24]. In fact, the results of pH stability revealed that MRPs and MRPL have a relative stability at pH 6.8, but some factors, such as the presence of trypsin and shaking at 100 r/min at 37 °C, accelerated the release collectively, since the outer lipid membranes were oxidized and disrupted and the odds of molecules colliding were increased. Overall, compared with free MRPs, MRPL exhibited a lower release in the simulated gastric and intestinal environment, indicating that microencapsulation with liposomes efficiently protects MRPs from being degraded and has the potential for site-specific delivery in biological systems [14]. 

As shown in Table 1, the release behaviors of MRPs and MRPL were evaluated by four drug-release kinetic models, including Zero order, First order, Higuchi and Korsmeyer–Peppas model. The release of MRPs and MRPL in SGF and SIF were best fitted into the Korsmeyer–Peppas model because of the highest linearity (R^2^). The constant n of the Korsmeyer–Peppas model reveals the dissolution mechanism: when n ≤ 0.43, the dissolution is dominant by Fick diffusion; when 0.43 ≤ n ≤ 0.85, the drug diffusion is integrated with erosion; when n ≥ 0.85, the diffusion is largely driven by erosion [21]. As shown in Figure 3c,d, the values of the exponent constant n of MRPs in SGF and SIF were 0.7451 and 0.6945, respectively, while those of MRPL were 1.0307 and 0.9096, respectively, which suggested that the release of MRPs in vitro was diffusion coupled with erosion, while that of MRPL was mainly erosion. Moreover, the K value of MRPL was smaller than that of MRPs, both in SGF and SIF, indicating that the release rate of MRPL was much slower due to the protective effect of liposome embedding [39]. 

### 3.4. Cytotoxicity to MKN-28 and HepG-2 Cells

#### 3.4.1. Cell Viability

As shown in Figure 4a,b, MRPs and MRPL all exhibited inhibitory effects on the survival of MKN-28 and HepG-2 cells with a dose-dependent manner. The IR of MRPL to MKN-28 cells significantly increased from 13.2% to 25.7%, while that of MRPs to MKN-28 cells increased from 9.9% to 21.3% when MRPs concentration was added from 0.08 mg/mL to 0.16 mg/mL. When the concentration doubled (0.32 mg/mL), the IR of MRPL (38.5%) to MKN-28 was nearly 6% higher than that of MRPs (32.5%).

Similarly, the IR of MRPs and MRPL to HepG-2 cells was dramatically increased to 12.5% and 15.2% at MRPs’ concentration of 0.08 mg/mL. Subsequently, the growth of IR had a slower trend than before as the concentration continued to rise, and the IR of MRPs and MRPL to HepG-2 cells were 21.2% and 23.8%, respectively, at the maximum concentration (0.32 mg/mL). 

Kurokawa et al. [5] reported that *Monascus* pigments possess great anticancer activity against murine 4T1 mammary carcinoma cell, and the cell viability reduced from approximately 50% to 5% when MP concentration increased from 300 to 400 μg/mL. Compared with free silibinin, an extraction from *Silybum marianum* seeds, the liposomal silibinin exhibited stronger cytotoxic activity against cancer cells [40]. The same trends are observed in propolis liposomes, which powerfully enhanced the cytotoxic effect on Hep-2 cells compared with the unencapsulated ones [26]. Because of the phospholipid membrane, liposome possesses characteristics of great biocompatibility and low toxicity [13]. Sun et al. [11] reported that the Caco-2 cells treated with anthocyanins-loaded nanoliposomes had lower cell viability compared with unencapsulated anthocyanins at the same concentration. Despite the anti-cancer cells’ effect of anthocyanins, AN greatly enhanced the uptake in Caco-2 cells because of the great affinity with the cell membrane. 

IC_50_ was employed to evaluate the half maximal inhibitory concentration of MRPs and MRPL on HepG-2 and MKN-28 cells, as shown in Figure 4c. The IC_50_ values of MRPs and MRPL in MKN-28 cells were 0.74 mg/mL and 0.57 mg/mL, respectively. Statistical analyses revealed that there are significant differences (*p* < 0.05) between MRPs and MRPL on IC_50_, indicating that embedding MRPs with lipid efficiently improved their inhibitory ability to MKN-28 cells. Moreover, there was no statistical difference (*p* > 0.05) in IC_50_ of HepG-2 cells between MRPs (1.93 mg/mL) and MRPL (1.92 mg/mL), suggesting that the cytotoxicity of HepG-2 cells is little improved by liposomal. Overall, MRPs after being wrapped by lipid displayed greater inhibitory activity to MKN-28 and HepG-2 cells compared with free MRPs under the same concentration, and the inhibitory effect on MKN-28 was better than HepG-2 cells.

#### 3.4.2. Cells Morphology

As shown in Figure 5a,d, the MKN-28 and HepG-2 cells without any treatment exhibited a state of rapid reproduction with high cell density and close arrangement. After 12 h of incubation with the same concentration of MRPs and MRPL, the MKN-28 cells began to take on a state of shrinkage with a dramatically decreasing cells number, shown in Figure 5b,c. Furthermore, the cells treated with MRPL had the worst integrity compared with free MRPs, which is well in line with the results of cell viability. Zheng et al. [41] reported that coix seeds displayed a great inhibitory effect on HEp2 cells by damaging the cell integrity after fermentation by *Monascus purpureus*. MP can significantly enhance the cytotoxicity against 4T1 cells by inducing cell apoptosis [5]. In addition, compared with the control cells, the number and integrity of HepG-2 cells was less and worse after being treated with MRPs and MRPL (Figure 5e,f). However, little obvious difference on HepG-2 cells morphology was observed between the treatment of MRPs and MRPL. 

## 4. Conclusions

Liposomes, as a promising carrier, have a potential application in pharmaceutical, cosmetics, and food industries. In this study, MRPL that was prepared by thin-film ultrasonic method was presented as spherical UV. The particle size, PDI, and zeta potential of MRPL contributed to better stability in adverse environmental conditions and simulated a gastrointestinal environment in vitro. This study provides information about the cytotoxicity of MRPL to some cancer cells, such as MKN-28 and HepG-2cells. Overall, liposomes embedding contributes to improving the stability of MRPs during processing and storage in the food industry. Their better stability and greater anticancer activity endow MRPL with a promising application in functional foods and pharmaceutical fields. 

## Figures and Tables

**Figure 1 foods-12-00447-f001:**
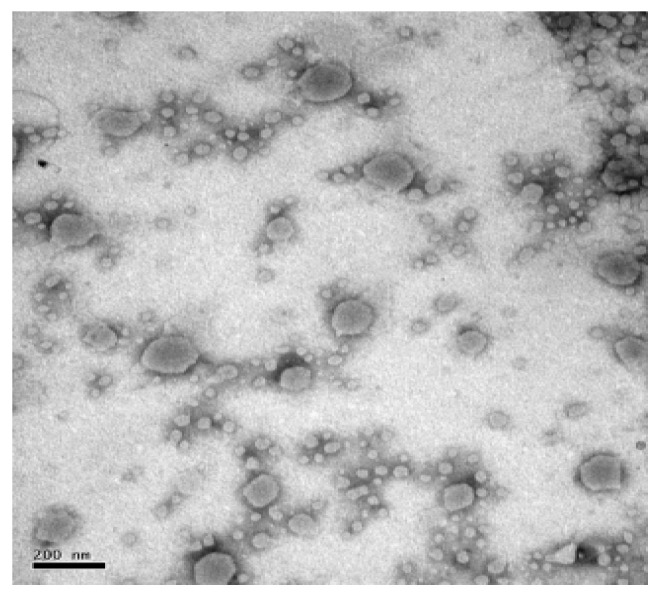
Transmission electron micrograph of MRPL.

**Figure 2 foods-12-00447-f002:**
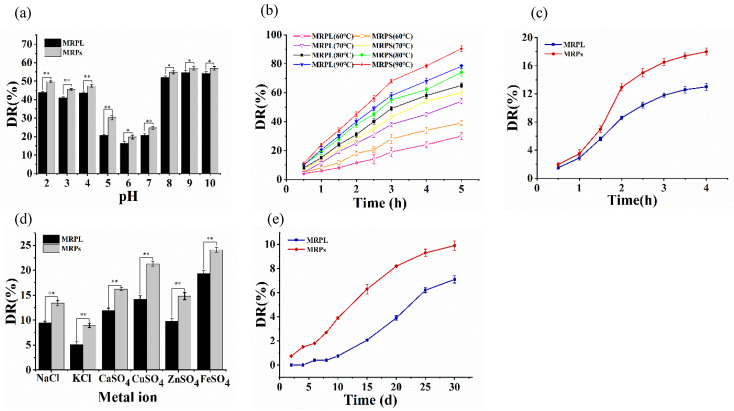
Stability of pH (**a**), temperature (**b**), light (**c**), metal ions (**d**) and storage (**e**) on free MRPs and MRPL. * significant, *p* < 0.05; ** extremely significant, *p* < 0.01.

**Figure 3 foods-12-00447-f003:**
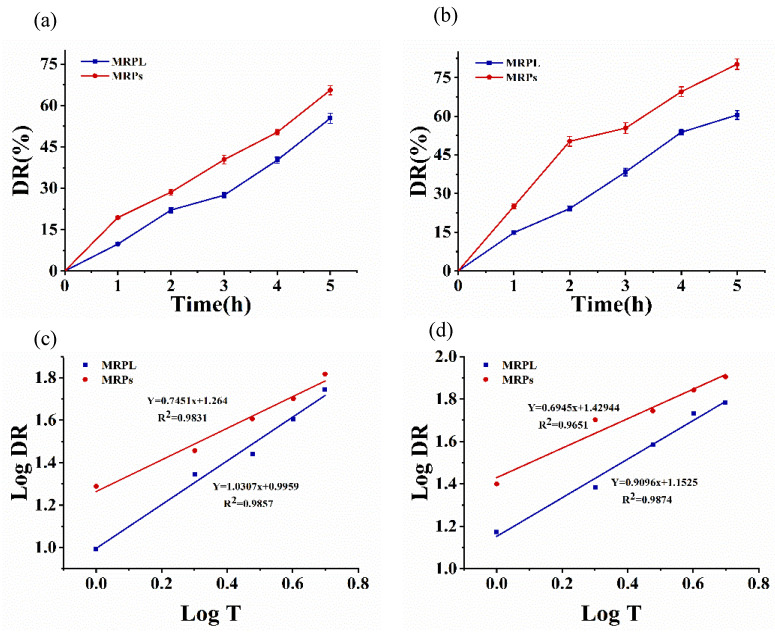
In vitro release of free MRPs and MRPL and Korsmeyer–Peppas kinetics. (**a**) Release in SGF of free MRPs and MRPL; (**b**) release in SIF of free MRPs and MRPL; (**c**) Korsmeyer–Peppas kinetics of release in SGF; (**d**) Korsmeyer–Peppas kinetics of release in SIF.

**Figure 4 foods-12-00447-f004:**
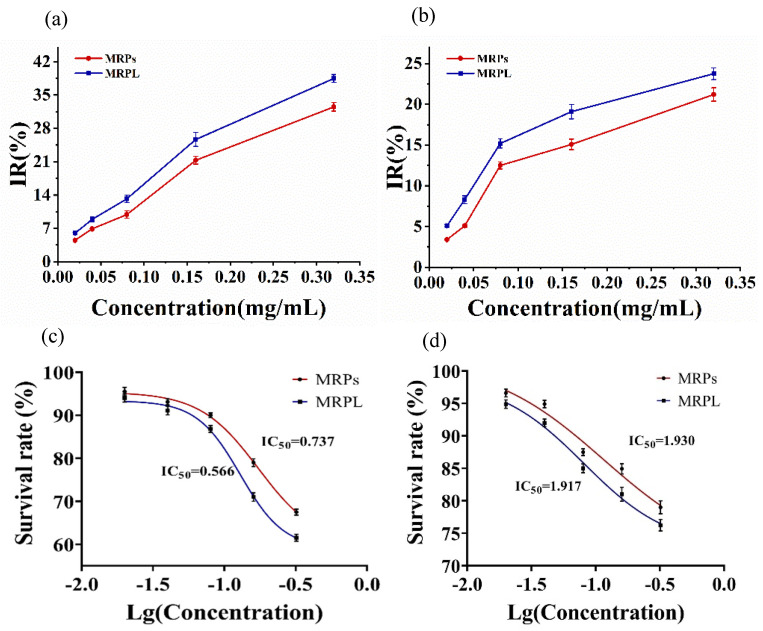
Cytotoxicity to MKN-28 and HepG-2 cells. (**a**) Inhibitory effect of free MRPs and MRPL on MKN-28 cells; (**b**) Inhibitory effect of free MRPs and MRPL on HepG-2 cells; (**c**) IC50 values of MRPs and MRPL on MKN-28 cells; (**d**) IC50 values of MRPs and MRPL on HepG-2 cells.

**Figure 5 foods-12-00447-f005:**
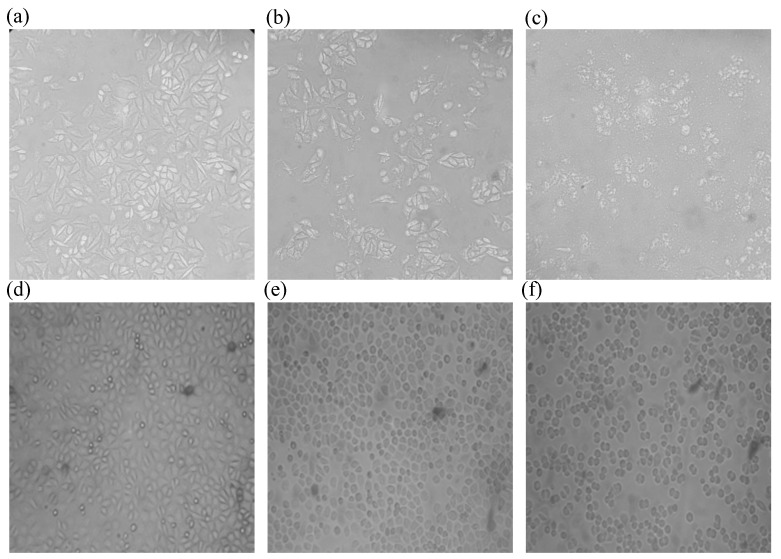
Effect of MRPs and MRPL on the morphology of MKN-28 (**a**–**c**) and HepG-2 cells (**d**–**f**). (**a**) Morphology of control MKN-28 cells; (**b**) Morphology of MKN-28 cells treated with MRPs; (**c**) Morphology of MKN-28 cells treated with MRPL; (**d**) Morphology of control HepG-2 cells; (**e**) Morphology of HepG-2 cells treated with MRPs; (**f**) Morphology of HepG-2 cells treated with MRPL.

**Table 1 foods-12-00447-t001:** Regression equation of four drug-release kinetic models in simulated gastrointestinal environment.

		Zero Order		First Order		Higuchi		Korsmeyer–Peppas		
		R^2^	K_0_(h^−1^)	R^2^	K_1_(h^−1^)	R^2^	K_H_(h^−1/2^)	R^2^	K_KP_(h^-n^)	n
SGF	MRPs	0.9884	0.1236	0.9703	0.1990	0.9453	0.2787	0.9897	0.1899	0.7451
	MRPL	0.9863	0.1067	0.9510	0.1526	0.8782	0.2323	0.9874	0.1014	1.0307
SIF	MRPs	0.9490	0.1541	0.9833	0.3115	0.9775	0.3607	0.9870	0.2623	0.6945
	MRPL	0.9916	0.1238	0.9834	0.1910	0.9215	0.2753	0.9917	0.1238	0.9096

## Data Availability

Data is contained within the article.

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
