# Peer review of "Monascus Red Pigment Liposomes: Microstructural Characteristics, Stability, and Anticancer Activity"

_foods, 2023, doi:10.3390/foods12030447_

Round 1
Reviewer 1 Report
The authors have presented Monascus Red Pigment loaded liposomes with a focus on structural characteristics, stability and anticancer activity. I have few concerns which require further clarification. Authors are encouraged to revise the manuscript based on the following points.
1. Overall, the manuscript requires extensive English language correction. It must be proof-read by a native English speaker for clarity and structural flow.
2. The main concern is that the authors did not mention how much MRP was loaded in the liposomes. This information is very important as many subsequent characterization steps are based on this critical information.
3. Some key information in various methods is missing. For example, during stability studies how did the sample absorbance was measured? was any pretreatment of MRPL samples done before UV analysis, such as centrifugation? Phospholipids and cholesterol usually interfere with the UV absorbance, did you consider UV interference? How did you handle this?
4. Once again in simulated GIT digestion, samples were directly analyzed using UV spectrophotometer. why?
5. Why there is no HPLC method for analysis? UV spectrophotometric analysis usually suffer with inference of excipients.
Reviewer 2 Report
In this manuscript, He and co-workers prepared monascus red pigment liposomes (MRPL) and studied the physical characteristics of liposomes (size, morphology, stability, etc.) and anti-cancer activities. Data seem to suggest MRPL, as a liposomal formulation, exhibited better stability and stronger inhibitory effect on MKN-28 cell lines, compared to monascus red pigment (MRP). Overall, the manuscript is well-written, and the experiments are detailed. However, the reviewer has the following concerns that the authors need to address:
1. The authors mentioned that MRPs have good solubility in water; however, when preparing liposomes, there were no procedures involving unencapsulated MRPs removal. For example, size-exclusive chromatography or dialysis techniques. The authors need to provide explanations why there were no such steps involved. In addition, it would be beneficial to calculate the content encapsulation efficiency for these liposomes containing MRP.
2. The reviewer wonders why the liposomes were prepared in distilled water instead of buffer. The reason behind this needs to be provided.
3. In Fig 1, the TEM and DLS results do not match each other. There are obviously a good number of small vesicles under microscopy, but the distribution curve on DLS was good, indicating uniform vesicles. Explanations need to be provided.
4. For temperature stability, the temperature ranges the authors tested were very high. Liposomes composed of lecithin and cholesterol usually don’t have that high phase transition temperatures (Tm); content leakage/decomposing is unavoidable above Tm. As a result, it is recommended to test the stability at lower temperatures (physiological range).
5. For the metal ion stability, the reviewer wonders if the type of anion makes a difference since both chloride and sulfate were used. Also, it would be interesting to test how Fe (III) affects stability.
6. It would benefit the readers if compound structures were added, including some other compounds the authors discussed multiple times in the manuscript. For example, in lines 299-301, it is hard to understand the sentence if the structure of anthocyanins is not shown.
Minor issues:
1. The word “Concentration” was misspelled in Figs 4a-b.
2. Scale bars in Fig 5 are missing.
3. MRPL in lines 26 and 27 were spelled as MPRL.
Reviewer 3 Report
The submitted article aimed to prepare MRPs liposomes by thin-film ultrasonic method to improve their stability. The stability of MRPL was evaluated in different pH, thermal, light, metal ion, storage, and in vitro simulated gastrointestinal digestion; the cytotoxicity of MRPL to MKN-28 and HepG-2 cells was also studied. This presented study is interesting and has made a systematical assessment of MRPL on microstructure characteristics, stability, and cell cytotoxicity. I have a few minor remarks for the authors:
Consider calculating the optimal conditions (pH, thermal, light, metal ion, storage) for the stability of MRPL.
Page 2, line 49, add a number of the reference in the brackets after Zhang et al. [9], Shaddel et al.[10], Caddeo et al. [15]
Page 3, line 120, the regression equation between the MRPs solution concentration and the absorbance should be numbered,
Page 6, line 256, correct to pH=5 and pH=6,
In all figures, move letters marking individual figures in the right bottom corner (a), (b), (c)…
Reviewer 4 Report
The manuscript foods-2106376 needs some improvements
1) Considering the large number of systems to deliver or stabilize bioactive compounds and the preparation methodologies, in the introduction part the advantages of using liposomes compared to other systems should be highlighted and references related to other studies reported.
2) Particles size: the authors report that particles size obtained by DLS and TEM are in line, but from the TEM micrograph in Figure 1 a large polydispersivity is evident. Could you explain it?
The authors did not consider the effect of pH, temperature, ionis and time on the particles size. It would be preferrable to add.
3) In the Figure 3, the release is reported and not the stability as reported in the figure caption.
4) The release data should be evaluated using mathematical models suitbale for the type of systems
5)Statistical analysis should be reported, it does not appear in any of the performed studies.
5) All the devices used should be reported
6) Cells morphology can be evaluated using specific dying
Round 2
Reviewer 1 Report
Authors have addressed all the queries and this paper can now be accepted for publication.
Reviewer 4 Report
The authors properly revised the manuscript. I suggest for pubblication